# Characterizing Tyrosinase Modulators from the Roots of *Angelica keiskei* Using Tyrosinase Inhibition Assay and UPLC-MS/MS as the Combinatorial Novel Approach

**DOI:** 10.3390/molecules24183297

**Published:** 2019-09-10

**Authors:** Jia-Hao Lee, Hui-Ching Mei, I-Chih Kuo, Tzong-Huei Lee, Yu-Hsin Chen, Ching-Kuo Lee

**Affiliations:** 1School of Pharmacy, Taipei Medical University, Taipei 11031, Taiwan; 2Department of Science Education, National Taipei University of Education, Taipei 10671, Taiwan; 3Department of Microbiology and Immunology, University of British Columbia, Vancouver, BC V6T 1Z3, Canada; 4Institute of Fisheries Science, National Taiwan University, Taipei 10617, Taiwan; 5Taichung District Agricultural Research and Extension Station, Council of Agriculture, Executive Yuan, Taichung 42081, Taiwan

**Keywords:** *Angelica keiskei*, tyrosinase, UPLC-MS/MS, chalcone, coumarins

## Abstract

In this study, an in vitro tyrosinase inhibition assay in combination with ultra performance liquid chromatography-orbitrap mass spectrometry (UPLC-orbitrap-MS) was developed for the rapid screening and identification of tyrosinase modulators from roots of *Angelica keiskei*. Of the 15 candidates considered, nine chalcones, xanthoangelols (**1**), B (**2**), D (**3**), E (**4**), G (**5**), H (**6**), 4-hydroxyderricin (**7**), xanthokeismin B (**8**) and (2*E*)-1-[4-hydroxy-2-(2-hydroxy-2-propanyl)-2,3-dihydro-1-benzofuran-7-yl]-3-(4-hydroxyphenyl)-2-propen-1-one (**9**), five coumarins, umbelliferone (**10**), selinidin (**11**), isopimpinellin (**12**), phellopterin (**13**) and xanthyletin (**14**), and one other compound, ashitabaol A (**15**), were distinguished between the test samples and the controls with statistical significance, and the structure of each compound was determined by comparing with in-house standards and the literature. Among these, six compounds, xanthoangelol (**1**), xanthoangelol D (**3**), xanthoangelol H (**6**), 4-hydroxyderricin (**7**), laserpitin (**16**) and isolaserpitin (**17**), were isolated from roots of *A. keiskei*. Of the compounds isolated, compounds **1**, **7** and **16** were subjected to tyrosinase inhibitory assay, and the IC_50_ values were 15.87 ± 1.21, 60.14 ± 2.29 and >100 μM, respectively. The present study indicated that the combination of in vitro tyrosinase inhibition assay coupled with UPLC-MS/MS could be widely applied to the rapid screening of active substances from various natural resources.

## 1. Introduction

Natural products have been used in the field of medicine and cosmetics for centuries. Their potential to treat various skin diseases and to improve skin condition is well-known. As ultraviolet (UV) radiation is a contributing cause for sunburns, wrinkles, premature aging, cancer and reduced immunity against infections, there is an increasing demand for products that provide protection against UV radiation [1]. Tyrosinase is a key enzyme that catalyzes the initial rate-limiting steps of melanin synthesis [2,3]. Abnormal and excessive melanin synthesis is the primary cause for skin disorders including melasma, senile lentigo, freckles and age spots [4]. As a result, finding novel and effective melanogenesis inhibitors has profound importance in controlling melanin production and pigmentation-related disorders [5]. In an attempt to find potent and safe tyrosinase inhibitors, this experiment evaluated the in vitro tyrosinase inhibitory activity of *Angelica keiskei*.

*A. keiskei* Koidzumi (Japanese name: ‘Ashitaba’, Umbelliferae) is a hardy perennial herb with exuberant vitality and multi-bioactive components. It originated in Hachijojima, Japan, and now mainly grows along the Pacific coast of Japan. *A. keiskei* is traditionally used as a diuretic, laxative, analeptic and galactagogue [6,7,8]. It is a dark green leafy vegetable that has been recognized as a medicinally important herb and cultivated in many Asian countries because of its health benefits. The stems and leaves have been consumed commercially as health foods and the roots have also been used as a food additive and medicine to alleviate pain and diabetic symptoms [9]. Various chalcones such as xanthoangelol, 4-hydroxyderricin and coumarins like xanthotoxin and laserpitin have been isolated and characterized from this plant [6,8].

Chalcones and coumarins are the main bioactive compounds in *A. keiskei*. Chalcones have been widely studied and are known to contain antioxidants [10], anticancer agents [11,12] and α-glucosidase inhibitors [13]. Coumarins, on the other hand, were proved to have antioxidants [14], antidepressants [15] and anticancer agents [8,16,17].

An ultra-performance liquid chromatography (UPLC) coupled with the high resolution MS/MS method is the advanced method for the identification and quantification of phenol compounds from plants and food [18]. The full scan analysis reveals the molecular weight (MW) of the unknown compounds, while tandem MS reveals aspects of the chemical structure of the precursor ion via fragmentation. Ion trap mass spectrometers can carry out sequential fragmentations of the precursor ion to form product ions. By using this method, unknown compounds are identified from the exact mass and MS/MS fragmentation. Since compounds are scanned separately and identified, the UPLC-MS/MS approach has increased sensitivity and provides more structural information based on fragmentation pattern of the analyte [19].

The aim of this study was to establish a rapid method by using UPLC-MS/MS to identify the components in *A. keiskei* that regulate tyrosinase activity via statistical analysis.

## 2. Results

### 2.1. The Tyrosinase Modulatory Effect of Angelica keiskei

According to the current understanding, *A. keiskei* has anti-inflammatory, antimicrobial and antihypertensive characteristics [14], while its effect on tyrosinase activity remains largely unknown [20,21]. With the lack of research in this field, our study focused on the analysis of tyrosinase activity in the presence of different concentrations of *A. keiskei* extracts from leaf and root parts. As shown in Figure 1, the preliminary results from our study showed that at the concentration of 500 μg/mL the tyrosinase activity remained unchanged and similar to blank when treated with *A. keiskei* leaves. On the other hand, a dosage-dependent inhibition of tyrosinase activity was observed when treated with *A. keiskei* roots. This trend of the inhibitory effect started at 200 μg/mL and increased with the higher concentration of *A. keiskei* roots. At the 1000 μg/mL level, conditions even exceeded the inhibitory effect achieved by 10 μM kojic acid (positive control). Since the leaf extract showed no evident change at the 500 μg/mL concentration, the subsequent experiments were performed using the root extract of *A. keiskei* with the purpose of further examining its modulatory effect on tyrosinase activity.

### 2.2. Screening Tyrosinase Modulator by UPLC-MS/MS

In order to investigate the compounds of *A. keiskei* roots that modulate the tyrosinase activity in a timely manner, we analyzed the samples using the simple LC-MS protocol optimized by our lab [22]. The base peak chromatograms (BPC) showed that under the ESI (+) MS mode in Figure 2, there were differences in signal intensities for the roots of *A. keiskei* treated with or without tyrosinase at the retention times of 7.93, 10.83, 10.84, 14.86, 14.88, 15.52, 18.56 and 19.04 min (Table 1). The base peak chromatograms (BPC) ESI (−) MS mode are shown in Figure 3; differences in signal intensities were observed at the retention times of 7.65, 10.08, 10.87, 13.20, 13.82, 14.32, 16.35 and 19.95 min (Table 2). Consequently, we suspect that components in the roots of *A. keiskei* might have a modulatory effect on tyrosinase activity.

### 2.3. Multivariate Analysis to Differentiate Processed A. keiskei

The highly complex UPLC-MS spectra are difficult to visually link to various components; therefore, multivariate data analyses were performed to comprehensively characterize the distinct composition of various metabolites from the untreated samples and the tyrosinase-treated samples. For each condition, there were three biological replicates of *A. keiskei* metabolites (*n* = 3). After performing UPLC-orbitrap-MS-based profiling on individual samples, each dataset was processed using SIEVE software to align and extract meanings. All datasets obtained from the two treatment groups were then analyzed with PCA, OPLS-DA and S-plot functions within the SIMCA-P software to find the candidates of interest.

Due to the similarity in compositions, the differences between untreated (blank) and tyrosinase-treated (test) samples were hard to identify based on the BPC chromatograms (shown in Figure 2 and Figure 3). Henceforth, a two-component PCA score plot of UPLC-MS data was utilized to depict general variations in components among the *A. keiskei* samples (Figure 4). According to the PCA scores plot in Figure 4, the distribution can be readily classified into two clusters, with the untreated (blank) and treated (test) samples clearly separated by principal component 1 (PC1) (Figure 4).

First, OPLS-DA was performed to compare untreated and tyrosinase-treated samples. An S-plot analysis was then used to select the critical variables that allowed differentiation. In the S-plot, each point represents *m*/*z*-RT pairs of molecules. To further reveal the tyrosinase inhibiting effect of *A. keiskei*, an OPLS-DA model was carried out between blank and test groups. The model fit well with R2Y value 0.996 and Q2 value 0.821 in positive mode, and R2Y value 1.0 and Q2 value 0.999 in negative mode. The score plot (Figure 4A,C) showed good separation, confirming the tyrosinase inhibiting effect. The corresponding S-plot was shown in Figure 4B, where coordinates in the lower-left quadrant were metabolites significantly increased in the blank group compared with the test group, while those in the upper-right quadrant represent the decreased ones. In Figure 4D, the components elevated in untreated samples were shown in the upper-right quadrant of the S-plot, while the lower-left quadrant revealed the components elevated in tyrosinase-treated samples.

### 2.4. Structural Characterization of Chalcones and Coumarines

To find the regulators for tyrosinase and the indicator component, the acetone extract was dissolved in acetone and coated in silica gel (1.5 g) and was then subjected to medium-pressure liquid chromatography (MPLC) and semi-preparative HPLC purification using the LC/MS-guided isolation approach. The process resulted in the successful isolation of six compounds, **1**, **3**, **6**, **7**, **16** and **17** (Figure 5). By using extensive NMR spectroscopic methods (1D and 2D-NMR), and LC/MS analysis to compare our results with previously reported spectroscopic values, the isolated compounds were determined to be xanthoangelol (**1**) [23], 4-hydroxyderricin (**7**) [24], xanthoangelol D (**3**) [25], xanthoangelol H (**6**) [26], laserpitin (**16**) [23] and isolaserpitin (**17**) [23] (Figure 5). These types of chalcones and coumarins are common amongst the *Angelica* species. In order to further confirm the effectiveness of the screening system, purified compounds, which in the S-plot are different between the blank and test groups, like the compounds **1**, **7** and **16**, were subjected to tyrosinase inhibition assay using kojic acid as the positive control. Results showed that both compounds **1** and **7** inhibited tyrosinase activity with IC_50_ values of 15.87 ± 1.21 μM and 60.14 ± 2.29 μM, respectively, whereas compound 16 had no inhibitory effect (IC_50_ > 100 μM) on tyrosinase. These findings corroborated the results found earlier via the screening system (Table 1, Table 2 and Table 3).

## 3. Materials and Methods

### 3.1. Reagents and Materials

The fresh *Angelica keiskei* was collected from Ali Mountain, Chiayi County of Taiwan, and authenticated by Dr. Yu-Hsin Chen (Taichung District Agricultural Research and Extension Station, Taichung, Taiwan). The voucher specimens (TMU-LCK-77) were deposited at the School of Pharmacy, Taipei Medical University, Taipei, Taiwan.

All the reagents including phosphate buffer (sodium phosphate monobasic (NaH_2_PO_4_), sodium phosphate dibasic (Na_2_HPO_4_)), kojic acid (positive control for the enzyme activity assay), dimethyl sulfoxide (DMSO) and acetone were purchased from Sigma Aldrich (St. Louis, MO, USA). Methanol and acetonitrile (all MS grade), on the other hand, were purchased from Macron Fine Chemicals™ (Radnor, PA, USA). The ultra-pure water was prepared with the Millipore-Q water purification system (Bedford, MA, USA).

### 3.2. Sample Preparation

The fresh *A. keiskei* was weighed, ground and precisely cut into thin blocks. The root (425 g) and leaf (490 g) specimens of *A. keiskei* were then purified and extracted three times using acetone in a 1:5 ratio to obtain 20.58 g and 18.11 g of crude extract, respectively.

#### 3.2.1. Crude Extract Preparation and Effective Compound Isolation

One gram of *A. keiskei* roots extracts was dissolved in acetone and coated in silica gel (1.5 g). The sample was subjected to medium-pressure liquid chromatography (MPLC, Isolera ONE, Uppsala, Sweden) on a silica SP column (Daiso gel, 50 µm, 10 g, column volume: 25 mL) with a fixed flow rate of 10 mL/min. The step-gradient of purification method was used to obtain the 0%, 20%, 30%, 40%, 60%, 80% and 100% fraction ethyl acetate solvents. A total volume of 100 mL was collected for each of the 6 ethyl acetate (EA) fractions (20 mL/tube). Tube numbers 12 to 15 were loaded on the same silica column for HPLC (250 × 10 mm, Luna 5µ Silica (2) 100 Å). The eluting solvent was ethyl acetate-hexane with a ratio of 20:80. At the flow rate of 4.0 mL/min, we were able to isolate 4-hydroxyderricin (**7**), xanthoangelol (**1**), laserpitin (**16**) and isolaserpitin (**17**). Similarly, HPLC was performed on tube numbers 18 to 23 on the same silica column. EA-hexane with a ratio of 40:60 was used as the eluting solvent. With the flow rate of 4.0 mL/min, we successfully isolated xanthoangelol D (**3**) and xanthoangelol H (**6**). All of the compounds were identified through HR-MS and NMR and their identities were confirmed via comparison with the literature values.

#### 3.2.2. Tyrosinase Assay

The L-tyrosine substrate and the mushroom tyrosinase were purchased from Sigma Aldrich. The test samples were prepared by first dissolving mushroom tyrosinase (250 U/mL) and l-Tyrosine (0.1 mg/mL) in DMSO and diluting the stock to different concentrations using phosphate buffer (66.7 mM at pH 6.8) to obtain a final DMSO concentration of 1%. The 96-well plate was divided into four groups: Blank control (BC), experimental control (EC), sample control (SC) and sample experiment (SE). All wells contained 40 μL of tyrosinase and had a total volume of 200 μL. On top of the enzyme solution, the BC group had phosphate buffer (160 μL) in the wells; EC contained phosphate buffer (80 μL) and l-Tyrosine (80 μL) in each well; SC wells consisted of phosphate buffer (120 μL) and sample solution (40 μL); SE wells, on the other hand, had phosphate buffer (40 μL), l-Tyrosine (80 μL) and sample solution (40 μL) in them. These assay mixtures were incubated at 37 °C for 30 min and measured at 475 nm using the microplate reader (Molecular Devices, CA, US). Percentages of tyrosinase activity were calculated using this formula:% Tyrosinase activity = [(SE − SC) ÷ (EC − BC)] × 100%(1)
Note:
SE = absorbance of sample experiment (with tyrosine)SC = absorbance of sample control (without tyrosine)EC = absorbance of experimental control (with tyrosine)BC = absorbance of blank control (without tyrosine)

### 3.3. UPLC-MS/MS Analysis

UPLC-MS/MS analyses were performed using a linear ion trap-orbitrap mass spectrometer (Orbitrap Elite; Thermo Fisher Scientific, Bremen, Germany) coupled with the online UPLC system (ACQUITY UPLC; Waters, Waters Corp., Manchester UK). The working solution was separated by an ACQUITY UPLC BEH C18 column (100 × 2.1 mm, 1.7 μm; Waters) at 40 °C. The flow rate of mobile phase A (ddH_2_O) and B (methanol) was 0.3 mL/min. The sample injected 5 μL. The gradient program was set as 20–30% phase B from 0–1 min and 30–100% phase B from 1–25 min. The column was washed with 100% phase B for 5 min before being re-equilibrated for 5 min. The mass spectrometer was equipped with an electrospray interface controlled by the Xcalibur software (version 2.0, Thermo Fisher Scientific, Bremen, Germany) with two types of operations: The positive ion and negative ion modes. The ESI source was set at these parameters: Spray voltage of 3.5 kV for the positive-ion mode and −3.2 kV for the negative-ion mode; capillary temperature was set at 360 °C and the source heater temperature was maintained at 350 °C. During the analysis, the mass spectrometer performed high-resolution (resolving power, r = 15,000) full scan cycles (*m*/*z* 100–1000). The analyte profiles were first made by the orbitrap; the MS2 spectra were then recorded in the centroid mode for the five most intense ions. The isolation width was set at the mass-to-charge ratio (*m*/*z*) of 0.2 and the higher-energy collisional dissociation (HCD) was performed at collision energies of 20, 30, 40 and 50 eV

### 3.4. In Vitro Tyrosinase Inhibition Assay by UPLC-MS/MS Analysis

The crude extract of *A. keiskei* root (15 mg) was dissolved in 75 μL of DMSO and 525 μL of 50 mM phosphate buffer (pH 6.8) to form the blank solution. Similarly, another 15 mg of the crude extract was dissolved in 75 μL DMSO and 425 μL of phosphate buffer solution (50 mM, pH 6.8) before 100 μL of tyrosinase (716 units/mL) was added as the test solution. Both solutions were stirred and incubated in the 37 °C water bath for 40 minutes in the dark. Subsequently, 900 μL of acetonitrile was added to each solution in order to terminate the enzyme reaction. Both test solutions with the final concentration of 10 mg/mL were filtered through the 0.22 μm filters. The working solutions were then analyzed by UPLC-MS/MS.

### 3.5. Mass Fragmentation Analyses and Compound Identifications

The MassLynx Mass Spectrometry software version 2.0.w.15 (Waters Corporation, Milford, MA, USA) was used for analyses. We extracted the mass fragment spectrum of each candidate in positive and negate mode, separately, and then transferred the mass fragment spectrum to the mass list. In the last step, via the MassFragment software in the MarkerLynx XS, structures were assigned to our observed fragment ions of small molecule compounds and, further, the reported structures were cross referenced with the literature on *A. keiskei.*

### 3.6. Statistics

The raw data were aligned and extracted using the SIEVE v2.2 application software from Thermo Fisher Scientific. The experimental target was metabolomics and the minimum intensity for the base peak was 1,000,000. The frames and threshold values were defined as 481,351 for the positive mode and 674,992 for the negative mode. SIMCA-P 13.0.3 software from MKS Umetrics (Umeå, Sweden) was then used to obtain OPLS-DA. The S-plots were also utilized for finding different candidates between blank and test groups.

## 4. Conclusions

Besides xanthoangelols (**1**), B (**2**), D (**3**), E (**4**), G (**5**), H (**6**), 4-hydroxyderricin (**7**), xanthokeismin B (**8**) and (2*E*)-1-[4-hydroxy-2-(2-hydroxy-2-propanyl)-2,3-dihydro-1-benzofuran-7-yl]-3-(4-hydroxyphenyl)-2-propen-1-one (**9**), umbelliferone (**10**), selinidin (**11**), isopimpinellin (**12**), phellopterin (**13**), xanthyletin (**14**) and ashitabaol A (15) in the positive and negative modes of UPLC-MS/MS, along with the differences between untreated and treated, revealed that there were nearly 15 different tyrosinase modulators found in these two groups. In addition to umbelliferone (**10**) and isopimpinellin (**12**), which are already known to inhibit tyrosinase activity, we also identified compounds **1** and **7** as active components with similar effects. This is the first research to investigate the tyrosinase-modifying effects of *A. keiskei*. We compared the compounds with the published literature and found xanthoangelols (**1**) [20], 4-hydroxyderricin (**7**) [20], umbelliferone (**10**) [27], isopimpinellin (**12**) [28] and phellopterin (**13**) [28] had been reported to have tyrosinase inhibition activity. The compounds from its roots demonstrate promising potential for treating hyperpigmentation and related disorders. Future research can focus on elucidating the in vivo effect and the optimal dosage.

With the aim of evaluating the effectiveness of the novel screening system, we examined the purified compounds to check the candidates from the base peak chromatograms and the tyrosinase inhibitory assay and found out that the results from the two experiments support each other. We therefore conclude this novel screening is highly accurate and effective. Furthermore, in comparison to the traditional protocols, our method reduces the sample and solvent volumes required for UPLC-MS/MS analysis. In addition, it allows researchers to quickly analyze and screen for more candidates with tyrosinase modulatory effects. This strategy can also be used for the rapid development of applications in other screening platforms.

## Figures and Tables

**Figure 1 molecules-24-03297-f001:**
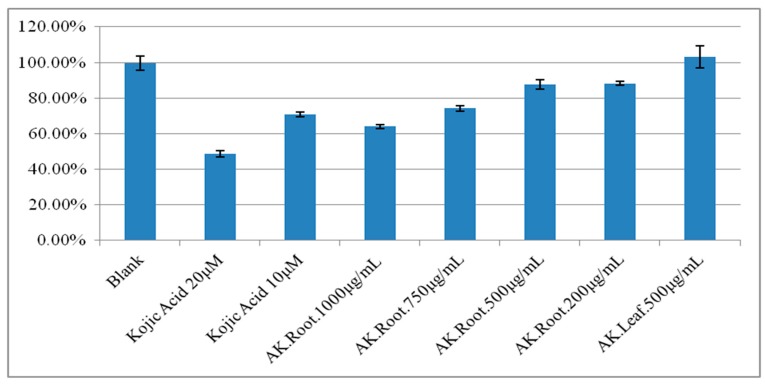
The comparative ratio of tyrosinase activity of the *A. keiskei* Leaf (*AK*. Leaf) at 500 μg/mL and roots (*AK*. Root) at 200, 500, 750 and 1000 μg/mL concentrations using kojic acid (10 and 20 μM) as the positive control.

**Figure 2 molecules-24-03297-f002:**
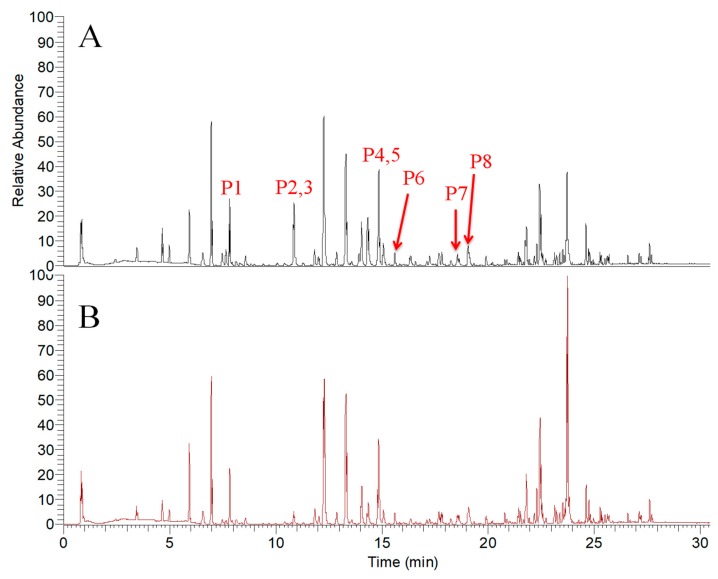
The base peak chromatograms (BPC) of untreated (**A**) and tyrosinase-treated samples (**B**) of *A. keiskei* in positive ion mode.

**Figure 3 molecules-24-03297-f003:**
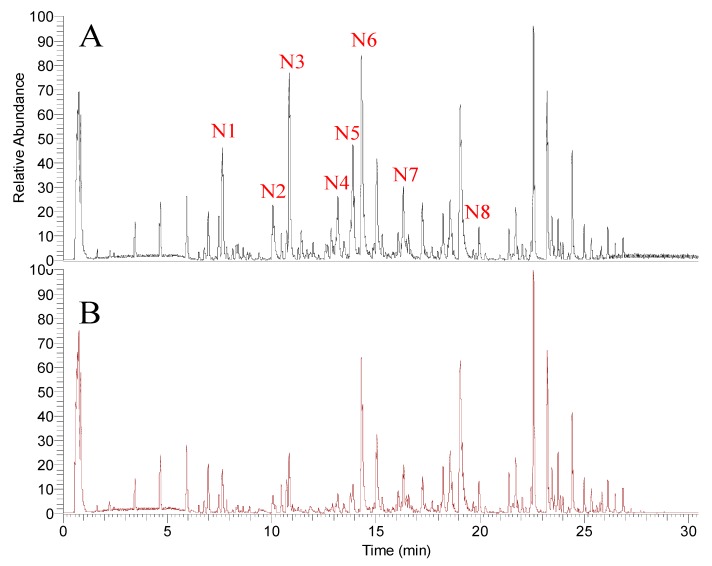
The base peak chromatograms (BPC) of untreated (**A**) and tyrosinase-treated samples (**B**) of *A. keiskei* in negative ion mode.

**Figure 4 molecules-24-03297-f004:**
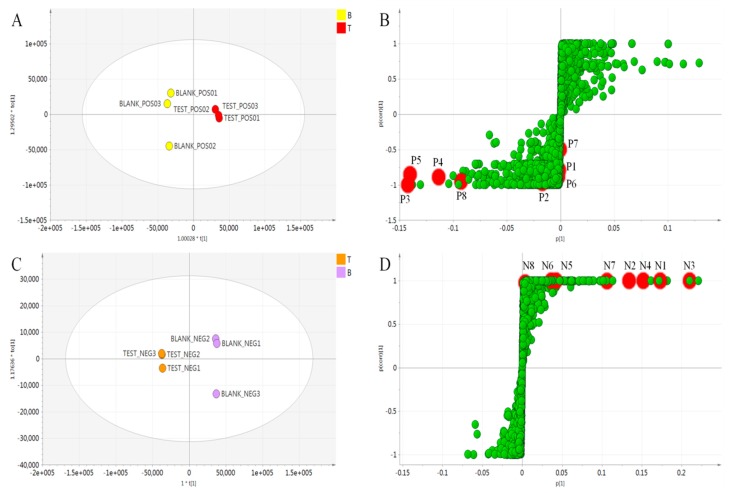
OPLS-DA score plot (**A**) and S-plot (**B**) of positive mode, OPLS-DA score plot (**C**) and S-plot (**D**) of negative mode, mass spectra obtained from untreated (blank) and tyrosinase-treated (test) groups.

**Figure 5 molecules-24-03297-f005:**
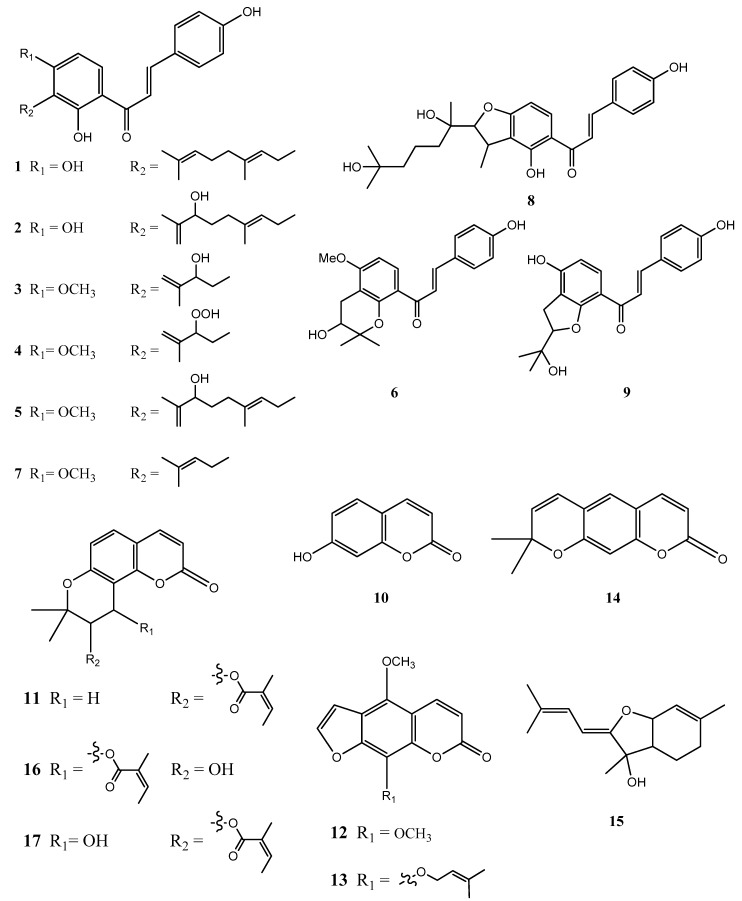
The chemical structure of 17 compounds in *Angelica keiskei* Koidzumi, xanthoangelol (**1**), B (**2**), D (**3**), E (**4**), G (**5**), H (**6**), 4-hydroxyderricin (**7**), xanthokeismin B (**8**), (2*E*)-1-[4-hydroxy-2-(2-hydroxy-2-propanyl)-2,3-dihydro-1-benzofuran-7-yl]-3-(4-hydroxyphenyl)-2-propen-1-one (**9**), umbelliferone (**10**), selinidin (**11**), isopimpinellin (**12**), phellopterin (**13**), xanthyletin (**14**), ashitabaol A (**15**), laserpitin (**16**) and isolaserpitin (**17**).

**Table 1 molecules-24-03297-t001:** Identification of the different amount of compounds from *A. keiskei* between the blank and test groups by UPLC-MS/MS in positive ion mode.

No.	t_R_ (min)	Measured *m*/*z* [M + H]^+^	Theoretical Formula [M + H]^+^	Error (ppm)	HCD (eV)	Fragment Ions (*m*/*z*) Abundance Rate (%)	Identification
P1	7.93	247.0605	C_13_H_11_O_5_	1.8	50	247.0608 (15) 232.0373 (40) 217.0137 (100)	Isopimpinellin
P2	10.8	163.0387	C_9_H_7_O_3_	−1.7	50	163.0389 (51) 133.0282 (72) 105.0332 (100)	Umbelliferone
P3	10.84	355.1542	C_21_H_23_O_5_	0.5	50	235.0959 (46) 217.0855 (45) 181.0493 (83) 163.0384 (100)	Xanthoangelol H
P4	14.86	229.0861	C_14_H_13_O_3_	0.9	50	229.0856 (100) 187.0386 (23) 175.0387 (50) 159.0438 (22)	Xanthyletin
P5	14.88	329.1384	C_19_H_21_O_5_	0.2	30	245.0807 (54) 227.0686 (100)	Selinidin
P6	15.52	235.1697	C_15_H_23_O_2_	1.7	50	235.1683 (100) 179.106 (82)	Ashitabaol A
P7	18.56	423.2170	C_26_H_31_O_5_	0.9	30	283.0945 (100) 181.0499 (23) 163.0375 (42) 131.5666 (18)	Xanthoangelol G
P8	19.04	339.1592	C_21_H_23_O_4_	0.2	40	283.0945 (32) 181.0484 (99) 163.0379 (100)	4-Hydroxyderricin

**Table 2 molecules-24-03297-t002:** Identification of the different amount of compounds from *A**. keiskei* between the blank and test groups by UPLC-MS/MS in negative ion mode.

No.	t_R_ (min)	Measured *m*/*z* [M + H]^+^	Theoretical Formula [M + H]^+^	Error (ppm)	HCD (eV)	Fragment Ions (*m*/*z*) Abundance Rate (%)	Identification
N1	7.65	369.1320	C_21_H_21_O_6_	−3.4	50	369.1389 (100) 297.0811 (1.3) 119.0517 (0.9)	Xanthoangelol E
N2	10.08	299.0905	C_17_H_15_O_5_	−2.9	40	299.0959 (100) 119.0516 (10)	Phellopterin
N3	10.87	353.1370	C_21_H_21_O_5_	−3.7	50	353.1436 (100) 239.1103 (2.4) 119.0518 (32)	Xanthoangelol H
N4	13.2	339.1216	C_20_H_19_O_5_	−3.1	50	339.1274 (100) 119.0517 (17)	(2*E*)-1-[4-hydroxy-2-(2-hydroxy-2-propanyl)-2,3-dihydro-1-benzofuran-7-yl]-3-(4-hydroxyphenyl)-2-propen-1-one
N5	13.82	439.1737	C_25_H_27_O_7_	−3.2	50	439.1779 (100) 319.1223 (3.4) 119.0525 (2.2)	Xanthokeismin B
N6	14.32	353.1373	C_21_H_21_O_5_	−2.9	50	353.1435 (100) 283.1014 (12) 119.0517 (32)	Xanthoangelol D
N7	16.35	407.1837	C_25_H_27_O_5_	−3.8	50	407.1913 (100) 287.1324 (5.5) 119.0517 (2.2) 243.142 (0.8)	Xanthoangelol B
N8	19.95	391.1892	C_25_H_27_O_4_	−3.0	50	391.1956 (100) 271.137 (11) 119.0518 (4)	Xanthoangelol

**Table 3 molecules-24-03297-t003:** Effect of kojic acid, 4-hydroderricin, xanthoangelol and laserpitin on mushroom tyrosinase activity.

Compounds	l-tyrosine IC_50_ (μM) (%) ^1^
Kojic acid ^2^	3.8 ± 0.2
Xathoangelol	15.87 ± 1.21
4-hydroyderricin	60.14 ± 2.29
Lasepitin	>100 μM (No inhibition at 100 μM)

^1^ Relative inhibitory activity, ^2^ Positive control.

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
