# Peer review of "Characterizing Tyrosinase Modulators from the Roots of Angelica keiskei Using Tyrosinase Inhibition Assay and UPLC-MS/MS as the Combinatorial Novel Approach"

_molecules, 2019, doi:10.3390/molecules24183297_

Round 1

Reviewer 1 Report

The authors submitted the manuscript entitled "Characterizing tyrosinase modulators from the roots of Angelica keiskei using tyrosinase inhibition assay and UPLC-MS/MS as the combinatorial novel approach. The manuscript was nicely written, although there are few spelling mistakes. 

Few comments:

Authors should focus on writing the introduction part, it is not up to date. Some of the recent reviews to be cited. For examples, Pillaiyar et al, J. Enzym Inhib. 2017; J.Med. Chem, 2018. 

It is good that they could identify the compounds are responsible for Tyrosinase inhibitory activity. However, authors should make very clear discussion about the new compounds identified for Tyr inhib. activity. 

What about the toxicity of compounds ?. Should be checked at least for new compounds. 

I would like to accept this manuscript after the corrections mentioned above. 

Author Response

Reviewer 1

Thank you for all your comments and suggestions. We corrected and reorganized our manuscript as recommended.

Authors should focus on writing the introduction part, it is not up to date. Some of the recent reviews to be cited. For examples, Pillaiyar et al, J. Enzym Inhib. 2017; J.Med. Chem, 2018.

Response 1: Thank you for the suggestions. We have included the indicated references in ref(3) and (5), and some illustration which helps to clarify the article has been added in Introduction. Please refer to the revised manuscript. (g. p 1 line 38, and p. 1 line 40 to 65).

Authors should make very clear discussion about the new compounds identified for Tyr inhib. Activity.

Response 2: Thank you for the suggestions. We made some discussion in Conclusions and added some references. Please refer to the revised manuscript. (g. p 10 line 245 to 247).

What about the toxicity of compounds ? Should be checked at least for new compounds.

Response 3:Thank you for the comments. We searched the literatures demonstrating the toxicity of compounds identified in our article. The organized descriptions and references are shown as follows.

No.

Compound name

Description about toxicity [Reference]

1

Xanthoangelol

No cell toxicity and apoptosis-inducing activity in human neuroblastoma (IMR-32) and leukemia (Jurkat) cells [1].

2

Xanthoangelol B

In Staphylococcus aureus, virulence factor expression is tightly regulated by a few master regulators, including the SaeRS TCS [2].aureus, virulence factor expression is tightly regulated by a few master regulators, including the SaeRS TCS [2].

3

Xanthoangelol DXanthoangelol D

No relative description was found in published papers.

4

Xanthoangelol EXanthoangelol E

No relative description was found in published papers.

5

Xanthoangelol GXanthoangelol G

No relative description was found in published papers.

6

Xanthoangelol HXanthoangelol H

No relative description was found in published papers.

7

4-hydroxyderricinhydroxyderricin

4-hydroxyderricin inhibited LPS-induced secretion of tumor necrosis factor-alpha (TNF-α) and expression of inducible NO synthase (iNOS) and cyclooxygenase-2 (COX-2) [3], and exhibited potent inhibitory activity on human DNA topoisomerase (Topo) II (IC50 = 21.9 µM) [4].hydroxyderricin inhibited LPS-induced secretion of tumor necrosis factor-alpha (TNF-α) and expression of inducible NO synthase (iNOS) and cyclooxygenase-2 (COX-2) [3], and exhibited potent inhibitory activity on human DNA topoisomerase (Topo) II (IC50 = 21.9 µM) [4].

8

Xanthokeismin BXanthokeismin B

No relative description was found in published papers.

9

(2E)-1-[4-hydroxy-2-(2-hydroxy-2-propanyl)-2,3-dihydro-1-benzofuran-7-yl]-3-(4-hydroxyphenyl)-2-propen-1-onehydroxy-2-(2-hydroxy-2-propanyl)-2,3-dihydro-1-benzofuran-7-yl]-3-(4-hydroxyphenyl)-2-propen-1-one

No relative description was found in published papers.

10

UmbelliferoneUmbelliferone

Umbelliferone was assayed against A-549 (Human Small Lung Carcinoma), HT-29 (Human Colon Carcinoma), HeLa (Human Cervical Carcinoma) RPMI (Human Nasal Septum Carcinoma) and HEp G2 (Human Liver Carcinoma) cell lines for cytotoxicity test on MTT assay. The cell viabilities were reduced by up to 62.5% at 1000 and 500 μg/ml concentrations compared to controls, while the test drug amount needed to inhibit cell growth by 50% was >1000 μg/ml against all cell lines [5].

11

Selinidin

No relative description was found in published papers.

12

Isopimpinellin

Imperatorin and isopimpinellin proved to be inhibitors of CYP 2B, whereas bergamottin and coriandrin inhibited the activity of CYP 1A1 and 1A2 in the liver. Furthermore, bergamottin inhibited the enzyme activity of CYP 3A [6].

Isopimpinellin (IC50 = 26.0 ± 5.7 mM) exhibited the highest activity against CEM/C1 (lymphoblastic leukemia) and HL-60/MX2 (MDR) cell lines [7].

13

Phellopterin

Phellopterin(IC50 = 8.0±4.0 mM)exhibited the highest activity against CEM/C1 (lymphoblastic leukemia) and HL-60/MX2 (MDR) cell lines [7].

14

Xanthyletin

No relative description was found in published papers.

15

Ashitabaol A

Ashitabaol A exhibited a diverse range of biological activities such as antifeedant, cytotoxic, anti-ulcer, and anti-inflammatory effects. Further, its total synthesis has been described [8].

16

Laserpitin

No relative description was found in published papers.

17

Isolaserpitin

No relative description was found in published papers.

Reference:

Tabata, K.; Motani, K.; Takayanagi, N.; Nishimura, R.; Asami, S.; Kimura, Y.; Ukiya, M.; Hasegawa, D.; Akihisa, T.; Suzuki, T., Xanthoangelol, a major chalcone constituent of Angelica keiskei, induces apoptosis in neuroblastoma and leukemia cells. Biological and Pharmaceutical Bulletin 2005, 28, (8), 1404-1407. Mizar, P.; Arya, R.; Kim, T.; Cha, S.; Ryu, K.-S.; Yeo, W.-S.; Bae, T.; Kim, D. W.; Park, K. H.; Kim, K. K., Total Synthesis of Xanthoangelol B and Its Various Fragments: Toward Inhibition of Virulence Factor Production of Staphylococcus aureus. Journal of medicinal chemistry 2018, 61, (23), 10473-10487. Yasuda, M.; Kawabata, K.; Miyashita, M.; Okumura, M.; Yamamoto, N.; Takahashi, M.; Ashida, H.; Ohigashi, H., Inhibitory effects of 4-hydroxyderricin and xanthoangelol on lipopolysaccharide-induced inflammatory responses in RAW264 macrophages. Journal of agricultural and food chemistry 2014, 62, (2), 462-467. Akihisa, T.; Kikuchi, T.; Nagai, H.; Ishii, K.; Tabata, K.; Suzuki, T., 4-Hydroxyderricin from Angelica keiskei roots induces caspase-dependent apoptotic cell death in HL60 human leukemia cells. Journal of oleo science 2011, 60, (2), 71-77. Rayar , A.; Manivannan , R., In vitro cytotoxicity activity of phytochemicals isolated from Coriandrum sativum linn in selected cell lines. Journal of Pharmaceutical and Biological Sciences 2015, 10, 38-49. Wen, Y. H.; Sahi, J.; Urda, E.; Kulkarni, S.; Rose, K.; Zheng, X.; Sinclair, J. F.; Cai, H.; Strom, S. C.; Kostrubsky, V. E., Effects of bergamottin on human and monkey drug-metabolizing enzymes in primary cultured hepatocytes. Drug metabolism and disposition 2002, 30, (9), 977-984. Kubrak, T.; Bogucka-Kocka, A.; Komsta, Ł.; Załuski, D.; Bogucki, J.; Galkowski, D.; Kaczmarczyk, R.; Feldo, M.; Cioch, M.; Kocki, J., Modulation of multidrug resistance gene expression by coumarin derivatives in human leukemic cells. Oxidative medicine and cellular longevity 2017, 2017. Aoki, N.; Ohta, S., Ashitabaol A, a new antioxidative sesquiterpenoid from seeds of Angelica keiskei. Tetrahedron Letters 2010, 51, (26), 3449-3450.

Reviewer 2 Report

The authors described a novel method for screening potential anti-tyrosinase compounds from plant source, and exemplified it with the identification of some modest inhibitors from Angelica keiskei, along the chemical isolation of a few constituents. Overall, the study is well conducted and technically rigorous, although the novelty factor is low. Therefore I would recommend publication in Molecules after the following minor points have been carefully addressed.

The names “Angelica” and “Angelica keiskei” should be written in italics (g. p 5 line 118, p. 6 lines 126 & 131, p. 7 line 134, etc.). Standard errors should be mentioned with the presented values, and the significant figures should be rounded accordingly (g. 15.87 ± 0.0X µM or 15.9 ± 0.X µM, p. 5 line 122).

Author Response

Thank the reviewers for the comments regarding our manuscript submission entitled:
Characterizing tyrosinase modulators from the roots of Angelica keiskei using tyrosinase inhibition assay and UPLC-MS/MS as the combinatorial novel approach. Below we have included a point-by-point response to each reviewer’s comment.
Reviewer 2
Thank you for all your comments and suggestions. We corrected our manuscript as recommended.
1. The names “Angelica” and “Angelica keiskei” should be written in italics (g. p 5 line 118, p. 6 lines 126 & 131, p. 7 line 134, etc.).
Response 1:Thank you for the notifications. We have corrected all the scientific names as italics type. Please refer to the revised manuscript.
2. Standard errors should be mentioned with the presented values, and the significant figures should be rounded accordingly (g. 15.87 ± 0.0X μM or 15.9 ± 0.X μM, p. 5 line 122).
Response 2: Thank you for the comments. The revised expression has been presented in the newly submitted manuscript. (g. p 1 line 27, p 5 line 137).
